# Brief communication: Small-scale geohazards cause significant and highly variable impacts on emotions

Evgenia Ilyinskaya [1], Vésteinn Snæbjarnarson [2,3], Hanne Krage Carlsen [4], Björn Oddsson [5]

[1]School of Earth and Environment, University of Leeds, Leeds, UK
[2]Miðeind ehf, Reykjavík, Iceland
[3]Department of Computer Science, University of Copenhagen
[4]Department of Public Health and Community Medicine, Institute of Medicine, Sahlgrenska Academy at University of Gothenburg, Gothenburg, Sweden
[5]Department of Civil Protection and Emergency Management, National Commissioner of the Icelandic Police, Reykjavik, Iceland

*Correspondence to*: Evgenia Ilyinskaya (e.ilyinskaya@leeds.ac.uk)

**Abstract.** The impact of geohazards on the mental health of the local populations is well recognized but understudied. We used natural language processing (NLP) of Twitter posts (n = 10,341) to analyse the sentiments expressed in relation to a pre-eruptive seismic unrest, and a subsequent volcanic eruption in Iceland 2019-2021. We show that although these geohazards were of small size and caused negligible material damage, they were associated with a measurable change in expressed emotions in the local populations. The seismic unrest was associated with predominantly negative sentiments (positive-to-negative sentiment ratio 1:1.3), but the eruption with predominantly positive (positive-to-negative sentiment ratio 1.4:1). We demonstrate a cost-effective tool for gauging public discourse that could be used in risk management.

## 1 Introduction

Social media posts have been successfully used for assessing geophysical aspects of volcanic and seismic events, for example, locating earthquakes using crowd-sourced information (Earle, 2010; Saraò et al., 2023; Steed et al., 2019; Wang et al., 2023); and science communication between researchers, and between researchers and citizens (Hicks, 2019; Lacassin et al., 2020; Watson et al., 2023). They have also been utilized for crisis and risk communication, rapid assessment of material damage, and aiding recovery efforts after several disasters including the Great Tohoku earthquake and tsunami; hurricanes (e.g. Harvey, Sandy, Eta and Iota); and major floods (e.g. (Bryan-Smith et al., 2023; Chatfield and Brajawidagda, 2012; Earle, 2010; Guan and Chen, 2014; Moghadas et al., 2023; Riddell and Fenner, 2021; Wang and Ye, 2018). Several recent studies have used social media posts for identifying challenges and inequalities in response, recovery and aid delivery following geodisasters (Kam et al., 2021; Nielsen et al., 2024; Olynk Widmar et al., 2022). The non-material impact of geohazards, including on the mental health of the local populations, is well recognized (Vo and Collier, 2013; Hlodversdottir et al., 2018; Becker et al., 2019; Gissurardóttir et al., 2019) but understudied. Studies are mostly done through interviews or clinical assessments, where

the results typically become available long after the event. Social media language and expressed sentiments can be used as indicators for public discourse and have been shown to be predictive of individuals' mental health state and its deterioration (Cha et al., 2022; Eichstaedt et al., 2018; Kelley and Gillan, 2022; Lan et al., 2024; Malko et al., 2023; Oltmanns et al., 2021; Recharla et al., 2024). Using artificial intelligence, such as natural language processing (NLP), it is possible to quickly process very large volumes of data for content and sentiment analysis (Bryan-Smith et al., 2023; Cha et al., 2022; Eichstaedt et al., 2018; He et al., 2022; Kam et al., 2021; Lan et al., 2024; Oltmanns et al., 2021; Park et al., 2015; Venkit et al., 2023).

Here we use NLP on a dataset collected on the social media platform Twitter (now known as X) to analyse the public views and expressed sentiments related to two types of globally common geohazards: a period of moderate seismic unrest and a small basaltic fissure eruption, using Iceland as the case study. The impacts of relatively small events are not well covered in literature, although they are much more common than large ones. Small eruptions (Volcanic Explosivity Index ≤2) account for ~80% of eruptions worldwide (Siebert et al., 2015). Pre-eruptive unrest is currently not considered in volcanic hazard and risk assessments beyond material damage to structures and has not been researched. There is a pressing need to understand the impacts of events of all sizes on the local populations, given that ~500 million people are living within 50 km distance from a volcano and rising numbers within 10 km (Freire et al., 2019).

## 2 Methods

The Reykjanes peninsula in Iceland provided a highly suitable natural laboratory. Between 2019 and 2021 this densely populated area (~260,000 people within 40 km radius) experienced two distinct and prolonged periods of geohazards: 15 months of elevated seismicity (Sigmundsson et al., 2022) (from here on termed 'seismic unrest period') that abruptly subsided and was followed by a basaltic fissure eruption that lasted 6 months (Halldórsson et al., 2022) (termed 'eruption period'). The seismic unrest period took place between December 2019 and March 2021, including several intense earthquake swarms, with the largest event of magnitude 5.6 (Sigmundsson et al., 2022). The eruption took place between 19 March and 19 September 2021 at Mt Fagradalsfjall and effused relatively small lava flows within uninhabited valleys. The material damage caused by the seismicity and the eruption was negligible and no physical harm was reported. We were able to focus our study on local residents rather than tourists by analysing social media posts written in Icelandic as the language is spoken predominantly by people living in Iceland. In addition, due to covid-19 restrictions for most of our study period, the number of people travelling internationally was at a record low in modern times.

Twitter was estimated to be used by 24% of Iceland's population at the time of study (Gallup, 2021). The main potential limitation of our approach is that views expressed on Twitter may not fully represent the views of people who chose to use different social media platforms or none at all; this can be explored in future research by including more than one social media platform. This would demonstrate the applicability of the method to other countries, where popularity of different social media platforms may differ.

## 2.1 Natural Language Processing

We performed sentiment analysis on tweets posted between 9 December 2019 and 31 December 2021 (n = 10,341) containing a fixed set of earthquake- and eruption-related keywords in Icelandic. Appendix A contains further details about the methods, including the full list of keywords used. A subset of 636 tweets was manually labelled as having 'negative sentiment', 'positive sentiment', or 'neutral statement' (see Table A2 in Appendix A for examples of labelled tweets). The rest of the dataset was labelled automatically into the same three categories by a pre-trained language model that was fine-tuned for classification using the manually labelled data. The model, bilingual in English and Icelandic (Snæbjarnarson and Einarsson, 2022), was first adapted for sentiment analysis using the English Stanford Sentiment Treebank (SST) dataset (Socher et al., 2013), as no explicit Icelandic sentiment analysis dataset exists, and this was the first time NLP sentiment analysis in Icelandic was attempted. The approach of first adapting the model using English-only data has been shown to enable cross-lingual transfer (Conneau et al. 2020, Pires et al. 2019). Finally, we trained on the Icelandic-only sentiment data. We acknowledge other ways to mix data but emphasise that our goal in this work was not to exhaustively compare these methods but simply evaluate one such methodology for the events we cover. This is a standard approach in NLP when data is used for transfer learning between a high-resource language and a low-resource language (Pfeiffer et al., 2020; Snæbjarnarson et al., 2023). Initial results showed that the model labelled earthquakes negatively and eruptions positively in sentences that should have been labelled as neutral. To mitigate this, we masked out all of the earthquake- and eruption-related keywords, both during model training and full dataset analysis. We hypothesise that this makes the model more robust to out-of-distribution settings, even though performance drops slightly on the evaluation split..Using a subset of tweets we manually verified that the model was labelling sentiments correctly as neutral when the keywords were masked out. The drop in model performance when masking was introduced was minor: there was a drop in accuracy from 71% to 69%, and F1 (the harmonic mean of the precision and recall) also dropped from 71 to 69. Full evaluation results are given in Appendix A. The model performance reached the 'benchmark' for good performance in Twitter sentiment analysis proposed by Zimbra et al., (2018). This demonstrates the potential of our method for broader use as sentiment analysis can be successfully achieved by NLP models bilingual in English and a local language without a need for a sentiment dataset in the local language.

## 3 Results and Discussion

The two geohazards in our case study evoked a measurable emotional response as indicated by the content of Twitter posts. Figure 1 is a timeseries spanning our study period and shows changes in the number of tweets with earthquake-related and eruption-related keywords (from here on termed 'earthquake-tweets' and 'eruption-tweets', respectively), and their sentiment labelling. Figure 1a shows the time-dependent changes in number of tweets, and the number of earthquakes by calendar week; Figure 1b shows the changes in number of tweets labelled as negative and positive, respectively, by week; and Figure 1c shows the changes in positive-to-negative sentiment ratio by week. The majority of the tweets containing earthquake- and/or eruption-related keywords (62%) were evaluated as containing a sentiment (30% negative and 32% positive). The remaining 38% were

neutral statements. Examples of tweets labelled as neutral, positive and negative are in Appendix A (Table A2). Previous work has shown that very large and/or destructive earthquakes, such as Great Tohoku 2011 (Vo and Collier, 2013), Canterbury 2010 (Becker et al., 2019), and Ridgecrest 2019 (Ruan et al., 2022) cause distress in the affected populations (measured using social media analysis by Vo and Collier, (2013) and Ruan et al., (2022); and 'traditional' interview methods by Becker et al., (2019)). We show here that earthquakes which are orders of magnitude smaller and cause no physical harm and negligible material damage can still evoke a significant emotional response (Figures 1b and 1c). We also show that the level of public interest is dependent on both the intensity of the seismicity (measured here as the number of earthquakes per week) and the proximity of the seismicity to densely populated areas. Figures 1a and 1b qualitatively show that the number of tweets increases in weeks with a high number of earthquakes, in particular when the earthquakes are located on the Reykjanes peninsula. To analyse this relationship quantitatively, we plotted the number of earthquake-tweets as a function of number of earthquakes, shown in Figure 2. For the densely populated Reykjanes peninsula, there is a correlation of $r^2 = 0.66$ (Figures 2a and 2b). The correlation disappeared when we consider the number of earthquakes in the rest of Iceland, which is relatively sparsely populated (Figures 2c and 2d).

The seismic unrest period was dominated by negative-sentiments with an average weekly positive/negative ratio 1:1.3. This finding agrees well with contemporary reports in the local media which described primarily negative experiences including anxiety (RÚV, 2021b) and disturbed sleep (RÚV, 2021a) caused by the earthquakes. The overall negative sentiments were likely caused by a combination of the physical discomfort of the ground shaking, and by the uncertainty about further development. It was already known that the seismicity was being caused by a magma intrusion but there was an uncertainty about whether, when, and especially where an eruption would happen; the areas considered to be at threat from lava flows included a town, and major infrastructure (power plants and tourism businesses). The seismic unrest was monitored closely and scientific interpretations were published continuously in the media. The constant reports will have been impossible to ignore and conflicting interpretations may have contributed to the anxiety and other negative emotions.

There was a statistically significant change ($p < 5e-05$) to predominantly positive sentiments associated with the start of the eruption (Figure 1b and 1c). The average weekly positive : negative ratio during the eruption was 1.4:1 compared to 1:1.3 prior. It is possible that the increase in positivity was even larger because the NLP evaluation consistently reported a smaller positive:negative ratio during the eruption period compared to the manual analysis (Figure 1c). To the best of our knowledge, this is the first time that an increase in positive attitudes associated with a start of an eruption is recorded among the local populations. We propose that the positive attitude is best explained by a combination of geophysical and societal factors. Many of the positive sentiments were expressing relief that an eruption would bring an end to the near-constant earthquakes, and to the associated uncertainty (examples of tweets: "*I want it to erupt so that the earthquakes stop*"; "*There is something absurd about being able to sleep soundly through the night now that there is an eruption in one's backyard*"). This agrees with previous studies on highly destructive events where positive statements were found associated with relief that the event is over (e.g. Great Tohoku 2011 earthquake; Vo and Collier, 2013), or messages of hope or pray for rescue and recovery (e.g. Hurricane Harvey; Zou et al., 2019). We also found that the Fagradalsfjall eruption directly evoked positive emotions in the local

populations, including joy and pleasure. Our findings suggest that the close proximity of the Fagradalsfjall eruption site to populated areas may have been an important factor that enhanced the positive attitudes, as it allowed more people to experience it first-hand (examples of tweets "*The eruption is so fantastic, even on my second visit <…> I adore seeing all kinds of people there enjoying the nature and being outdoors*"). The site was open to the general public and was within one hour's drive from the capital city, followed by a one hour hike. The public started arriving in large numbers within hours of the eruption and the total number of visits via the hiking trails equated 310,000 (Volcanic eruption in Geldingadalir, 2021). It is possible that the social and physical isolation caused by the covid-19 pandemic further enhanced the positive experience, as a visit to the eruption site provided both a distraction and an opportunity to socialize and exercise in a relatively covid-safe outdoor setting. Furthermore, the eruption had an unprecedented live coverage through multiple high-quality webcams, and social media feeds (Wadsworth et al., 2022), allowing participation and enjoyment of people who could not travel in person, and thereby potentially kicking off a new chapter of 'remote' volcano tourism. "*It is midday and I have done nothing except watching the eruption* [via live streams]. *So beautiful!*"). Previous studies focusing on tourists in volcanic areas e.g. (Benediktsson et al., 2011; Davis et al., 2013; Donovan, 2018) have reported overwhelmingly positive attitudes, but given their very limited participant number and type, it was unknown how the findings related to the general public in local communities. Our results show that eruptions that do not directly endanger lives and infrastructure may cause a measurable increase in positive attitudes on population-wide level, and furthermore suggest that given the opportunity, people across different ages and physical abilities are keen to experience volcanic activity up close.

**4 Conclusions**

Our findings are important for risk assessment and management because we show that even small-sized geohazards without significant material damage can cause a measurable change in expressed sentiments in the local populations, which in turn, may indicate an impact on people's mental health. Pre-eruptive unrest is currently a somewhat of a 'forgotten' volcanic hazard, the public health impacts of which have not been researched. Furthermore, it is not included in educational and scientific resources such as the Encyclopedia of Volcanoes (Rymer, 2015) and the VolFilm series (Global Volcanism Program | Video Collections | VOLFilms Collection, 2024). We show that pre-eruptive events (here, the seismic unrest) can potentially be more detrimental to mental wellbeing than the actual eruption and should be considered in studies of impacts. In general, the impacts of living with uncertainty due to expectant, but not yet materialised geohazards need further attention from the research and disaster risk reduction (DRR) communities.

Incorporating sentiment analysis of crowd-sourced information, such as social media posts, into local risk management has the potential for immediate and longer-term benefits. While our method does not provide direct measures of the mental health state and impacts, and is not intended to replace more formal investigations, it may be used to quickly gauge whether communities are under stress and may require additional surveying and/or resources. Alternatively, knowing that the public

views an eruption (or other natural phenomena) as a generally enjoyable and attractive event may assist the risk managers in choosing the most effective approach to achieve compliance should they need to restrict the site access.

Our study provides a valuable contribution to longitudinal studies of mental health impacts in the local populations. The volcano-seismic events on Reykjanes are still ongoing at the time of writing (June 2024) and are considered likely to continue for years or potentially decades. Long-duration, dynamically evolving volcanic events are common worldwide, for example at Kīlauea volcano in Hawaii. The method we demonstrate allows the capture and analysis of sentiments experienced and expressed *in situ*, which cannot be reliably reproduced by retrospective investigations. The contemporary meaning of people's lived experiences is essential but ephemeral, and the ability to recall fades and changes over time. With the literal loss of familiar landscapes, and homes, people also lose connection to their memories (Árnason and Hafsteinsson, 2023). Further studies could focus on designing methodologies for quantifying emotional impacts, mental health outcomes and risk perception changes using social media data in the Reykjanes case study, and elsewhere.

Finally, in our quest to reduce the risk posed by geohazards – in this case, a small eruption - we should not dismiss the potential mental health benefits from allowing people to experience them where possible, even though the benefits will be difficult to quantify and weigh up against the (often more obvious) risks.

**Data and code availability**

Historical tweets can be obtained from Twitter API through an academic research access. The code used here for downloading the tweets is a python package TwitterAPI (source code available at https://github.com/geduldig/TwitterAPI). The datasets and code generated in this study are available in an open-access repository https://github.com/vesteinn/fagradalsfjall_eruption_sentiment.

**Author contributions**

EI and HKC conceived the study idea. VS obtained and processed the Twitter data and led the NLP analysis with input from EI. EI led the manual Twitter data analysis. EI led the overall manuscript writing with input from all authors.

**Acknowledgements**

Twitter data was made available by academic API access of Twitter. EI acknowledges funding from NERC COMET+. VS acknowledges support by the Pioneer Centre for AI, DNRF grant number P1.

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

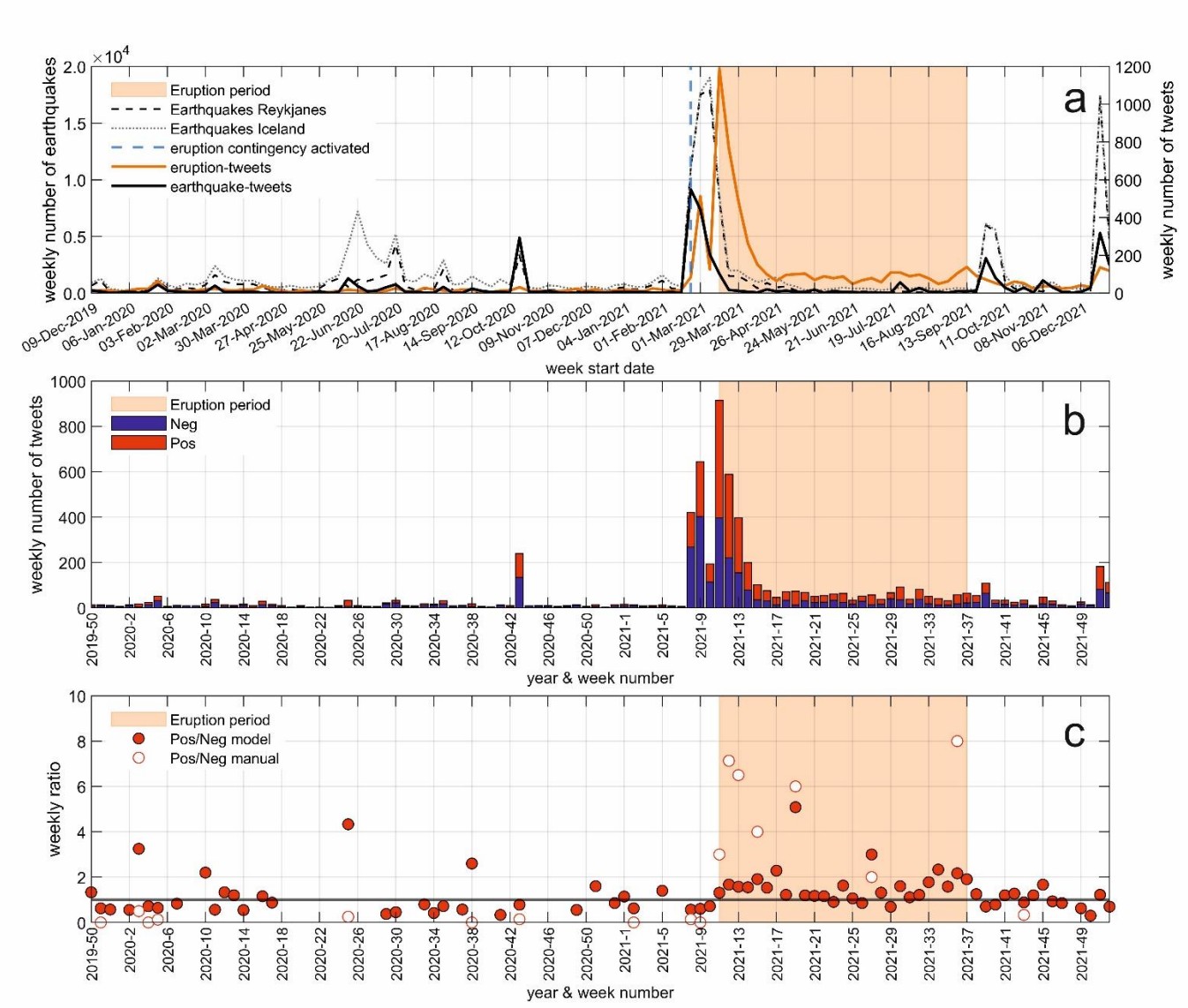

**Figure 1: Timeseries plot of Twitter data (*n* of Tweets = 10,341) during the pre-eruptive seismic unrest period (December 2019 – March 2021) and the eruption period (March – September 2021, highlighted with orange). The tweets were selected by a set of earthquake- and eruption- related keywords in Icelandic listed in Appendix A. a) The weekly number of earthquake- and eruption-**

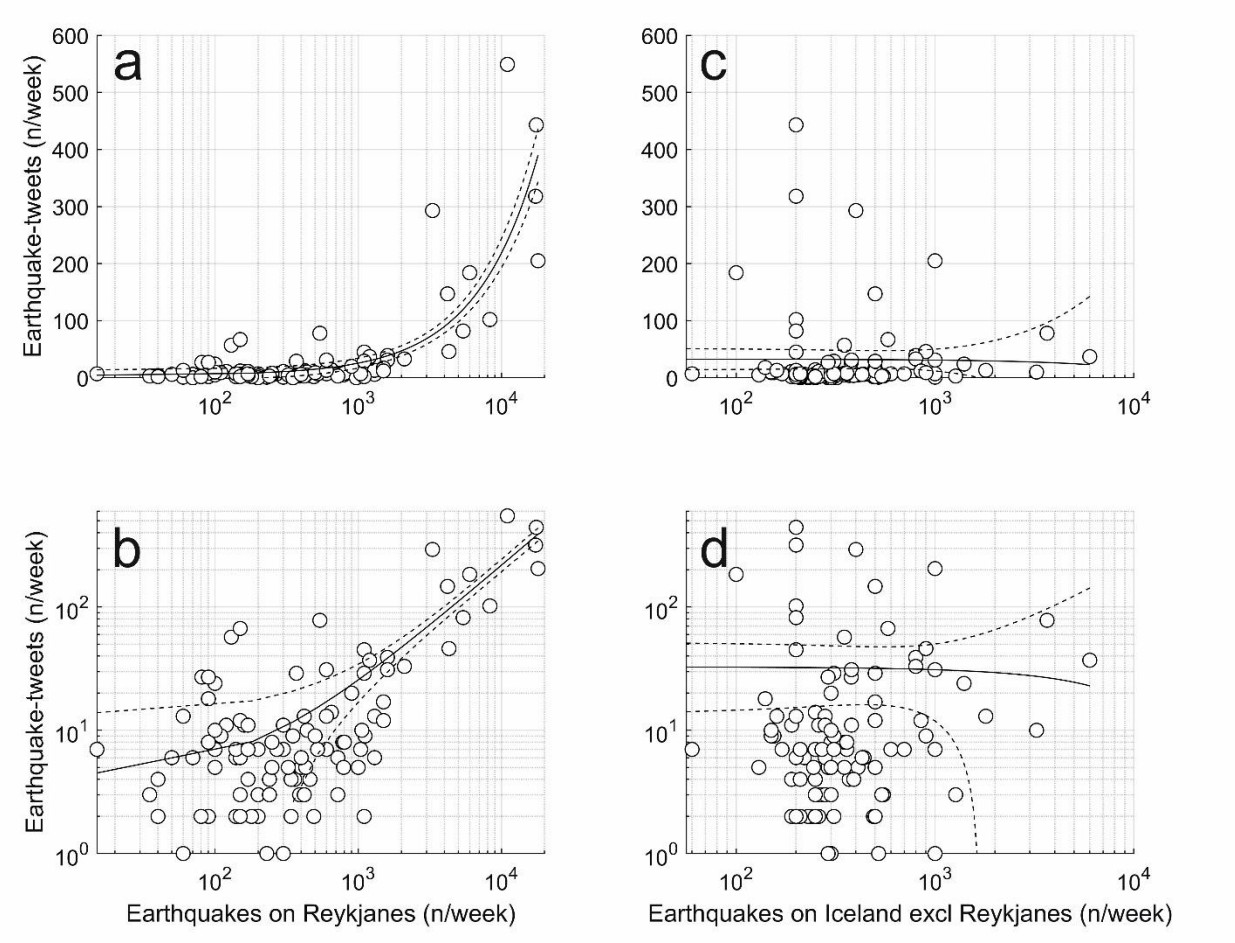

related tweets together with the weekly number of earthquakes. Number of weekly earthquakes is shown separately for Reykjanes peninsula and Iceland as a whole. The magnitude of earthquakes ranged from Mw 0 to 5.6. b) The weekly number of tweets evaluated as expressing positive ('pos') or negative ('neg') sentiments by the NLP model. c) The average positive/negative ('pos/neg') weekly ratio, as evaluated by the NLP model in the whole dataset, and manually in the data subset. The data shown in c) include only weeks where the total number of tweets was > 10 to avoid bias introduced by very low numbers.

**Figure 2: Scatter plot showing the number of earthquake-related tweets as a function of the number of earthquakes on a) the Reykjanes peninsula (earthquake magnitude range Mw 0 – 5.6), r² = 0.69 and b) the rest of Iceland, excluding Reykjanes peninsula (earthquake magnitude range Mw 0 – 4.8), no statistically significant relationship. The solid line is a linear regression model fit and the dotted lines are the 5% significance level**

**Appendix A**

**Theory and previous work on tweet-sentiment analysis**

Neural language models using the Transformer architecture (Vaswani et al., 2017) have had a great impact in the field of natural language processing (NLP). These models are trained using representation learning to embed words (create vector representations for the words and their parts) for use in tasks such as machine translation, question answering and sentiment analysis. Such models are first trained in an unsupervised manner on raw text to maximize embedding quality across all available contexts. The models, e.g. BERT (Devlin et al., 2019) and T5 (Kale and Rastogi, 2021), can then be adapted using fine-tuning on much smaller labeled datasets for a variety of classification tasks that depend on the context of text in natural language.

Sentiment analysis is a common area of research in NLP with connections to hate speech analysis and stance detection(Maas et al., 2019; Mohammad et al., 2016; Socher et al., 2013) Here we use it in a traditional positive, negative and neutral classification setting but with regards to the geohazards surrounding the volcanic eruption in Fagradalsfjall in 2021.

**Extended methods**

Having gained academic access to historical tweets, we use the python package TwitterAPI (source code available at https://github.com/geduldig/TwitterAPI) to download tweets posted between 9 December 2019 and 31 December 2021. The tweets are filtered for a fixed set of earthquake- and eruption-related keywords in Icelandic (details on keywords below). Duplicates, including retweets, were removed from the data. In total 10,341 individual tweets matched our criteria and were used for further analysis. We set the keywords to include all correct grammatical variations of 'eruption' and 'earthquake' in Icelandic, including plural. Spelling errors were not dealt with, this could be addressed with Levenshtein-distance but would also increases the risk of false positives. There are common words in Icelandic that have only small spelling differences with *gos* (e.g. gas) and these would have increased the amount of false positives. Another approach could have been to automatically correct the tweets but developed methods target editorial text rather than social media posts (Ingólfsdóttir et al., 2023). We chose this approach because the main keywords, *gos* and *skjálfti* (eruption and earthquake) and their grammatical variations, are short and easy-to-spell words in Icelandic.

For keywords, tweets were filtered with the morphological variations of eruption and earthquake in Icelandic, along with the emoji for volcano. The full list used is:

Eruption-related: ' 🌋 ', 'gos', 'gosið', 'gosinu', 'gossins', 'eldgos', 'eldgosið', 'eldgosi', 'eldgoss', 'gjósa', 'gaus'.

Earthquake-related: 'skjálfti', 'skjálfta', 'skjálftar', 'skjálftum', 'skjálftarnir', 'skjálftana', 'skjálftanum', 'skjálftann', 'skjálftan', 'skjálftans', 'skjálftinn'.

A subset of this dataset was manually annotated by sentiment into negative, positive or neutral. 224 tweets were labeled as negative, 241 as positive and 171 as neutral. Tweets were selected for manual annotation semi-randomly; outside of the eruption period, we randomly selected a subset of tweets in periods with either highly elevated seismicity or relatively low

seismicity levels. During the 6-month-long eruption period, we randomly selected a subset of tweets from each of the 3 first weeks of the eruption when eruption-related tweet frequency was at the highest level. We then randomly selected tweets in

eruption weeks 5, 9 17 and 26 to cover different time periods of the eruption.

To label the other tweets automatically an already adapted language model was fine-tuned on the manually annotated data for use in labeling as follows: The data was split randomly into a train and validation set of 493 and 121 tweets respectively. A pre-trained Icelandic-English language model (Snæbjarnarson and Einarsson, 2022) was first adapted for binary sentiment analysis using the English Stanford Sentiment Treebank (SST) dataset (Socher et al., 2013). The original bilingual model was

trained using Fairseq (Ott et al., 2019), from Facebook AI research, but then ported to the Transformers (Wolf et al., 2020) library and adapted for sentiment analysis. While no explicit Icelandic sentiment analysis dataset exists, the model seems to behave well in general sentiment analysis for Icelandic due to its bilingual pre-training, this behavior has been documented well before and is referred to as transfer-learning (Ruder et al., 2019). The model was then further fine-tuned on the training data for five full iterations (epochs) using a learning rate of 2e-5 and a batch size of 8. As the initial SST fine-tuned model was

only trained to classify positive or negative sentiment the output layer of the neural network was modified to provide three classes with the new classification node initialized by averaging over the prior two. Initial results showed that the model found that earthquakes were negative and eruptions positive in sentences that should have been labeled as neutral. To mitigate this, commonly seen short-cut to labeling (Geirhos et al., 2020), we masked out all of the earthquake- and eruption- related keywords, both during model training and actual evaluation. Using a subset of tweets we

manually verified that the model was labeling sentiments correctly as neutral when the keywords were masked out. The drop in performance when masking was minor: there was a drop in accuracy from 71% to 69%, and F1 (the harmonic mean of the precision and recall) also droped from 71 to 69. Evaluation results are shown in Table A1.

It is worth noting that the representations one can extract from pre-trained foundation models are limited to the relations that can be inferred from their training data. If these do not sufficiently cover the style or terminology one is interested in, then the

representations will not be robust. This can be particularly pronounced in the case of idiomatic expressions for low-resource languages when using multilingual models. As we targeted Icelandic, we used a pre-trained model trained on only Icelandic and English to mitigate these issues. If the method is to be followed for other languages, a limiting factor is that the mentioned resources need to be present in some form to cover a given language, i.e., pre-training data or pre-trained models, along with sufficient labelled sentiment data. In these cases, recent advances in general-purpose large-language models (see, e.g. (OpenAI

et al., 2024)) may be of value in both direct evaluation or pseudo-labelling training data in low-resource settings. Furthermore, fine-tuning these models using the labelled data can be seen as a way to improve these representations and their decision boundaries. Finally, as improved models and datasets become available, we can expect higher accuracy and lower uncertainty when these are applied to the data we have collected.

| Keywords masked | Accuracy | F1 | Recall | Precision |
|---|---|---|---|---|
| No | 71.4% | 71.3% | 72.9% | 71.6% |
| Yes | 69.0% | 68.8% | 70.5% | 68.9% |

**Table A1: Evaluation results for models trained on the collected tweets**

Examples of tweets labelled as neutral, positive and negative are shown in Table A2. Table A2 also shows whether the expressed sentiment was straightforward or ambiguous to label. Use of humor and sarcasm were common reasons for
ambiguous sentiment labelling.

The datasets and code are available in an open-access repository https://github.com/vesteinn/fagradalsfjall_eruption_sentiment for reproducibility and further use by the community.

The key NLP contributions are; a new dataset for Icelandic sentiment analysis of tweets about geological events, a fine-tuned Icelandic sentiment model by means of transfer learning and manually curated data available as open-access, and an evaluation
of masking to prevent shortcut learning in sentiment analysis.

| Label | Label certainty | Earthquake (E) or Volcano (V) keywords | Timestamp | Original tweet (Icelandic) | English translation |
|---|---|---|---|---|---|
| Neutral | Clear | E | 21/06/2020 08:15 | Stóru skjálftarnir í gær voru 5,4 og 5,6 á stærð | The large earthquakes yesterday were magnitude 5.4 and 5.6 |
| Negative | Clear | E | 26/02/2021 12:55 | Mér var kippt undan fallandi vegg í 17.júní skjálftanum 2000 og fékk bókahillu yfir mig í skjálftanum 3 dögum seinna. Var 5 ára Er búin að titra og | I got rescued from a collapsing wall in the 17 June 2000 earthquake and was hit by a bookshelf in the earthquake 3 days later. I was 5 years |

| | | | | |
|---|---|---|---|---|
| | | | | gráta við hvern skjálfta síðustu daga, áföll eru áhugaverður andskoti. | old. I have been shaking and crying after each earthquake in the last days, traumas are an interesting demon. |
| Negative | Clear | E | 24/02/2021 12:18 | Að sitja heima hjá mér í panikki eftir stóran skjálfta og vita ekki hvort ég sé að upplifa eftirskjálfta, kvíðaskjálfta í líkamanum eða hvort barnið í leginu á mér sé á svona mikilli hreyfingu að ég hrisstist er með top 5 því skrítnasta sem ég hef upplifað. | Sitting at home in a panic after a large earthquake and not knowing if I'm experiencing an aftershock, anxiety shakes in my body, or whether the baby in my uterus is moving so much that I'm shaking is among the top 5 weirdest experiences of my life. |
| Negative | Clear | V | 19/03/2021 22:14 | Er ég ein um að finnast ekkert spennandi við eldgos á þessum slóðum og kvíða því frekar hversu stórt það verður og hversu lengi og mun það hafa áhrif á samgöngur og og og smá kvíði hérna. | Am I the only one who is not excited about an eruption in this location and feeling anxious over how large it will be and for how long, and if it will impact transport and and and a bit anxious over here |
| Negative | Clear | V | 19/03/2021 23:19 | Hve vitlaust getur fólk verið.Gos og allir | How stupid can people be. [There is] an |

| | | | | keyra í fangið á gosinu og ganga síðasta spilinn upp á gígjinn | eruption and everyone is driving straight towards it and walking up to the crater |
|---|---|---|---|---|---|
| Positive | Clear | E | 20/10/2020 14:02 | Ég var með fulla stofu af 9. bekkingum á þessum stað í námsefninu þegar skjálftinn reið yfir. Nemendur hlógu og skríktu eins og í tívolítæki og vildu meira. Það sýnishornið. | I had a full classroom of 9th graders at this point in the curriculum when the earthquake struck. The pupils laughed and squealed like on a funfair ride and wanted more. What a point of view. |
| Positive | Clear | E | 24/02/2021 10:53 | Finnst ég vera í minnihlutahópi sem hamfaraperri. Sit í sófa í vinnunni, óvinnufær útaf tryllings spennu yfir næstu skjálftum! Sendi pabba skilaboð (fellow hamfaraperri), hann er í Hörpu, þar nötrar allt gler og Sinfóæfingu var hætt! Sorry ef þetta triggerar eh en þvílík veisla!! | I feel that I'm in a minority group as a disaster fetishist. Sitting on the sofa at work, unable to work because of extreme excitement over earthquakes! Messaging my dad (a fellow disaster fetishist), he is at Harpa [concert hall], all glass panes are shaking there and the Symphony stopped their practice! Sorry if |

| | | | | | this is triggering but what a feast!! |
|---|---|---|---|---|---|
| Positive | Clear | V | 12/04/2021 23:46 | Gosið er svo mikið æði, líka í annað skipti. Ótrúlegt að sjá gígana fjóra. Svo mikil upplifun. Dýrka líka að sjá alls konar fólk þarna að njóta útivistar og náttúru. | The eruption is wonderful, including on a second visit. Unbelievable to see the four craters. Such a huge experience. I also adore seeing all kinds of people there enjoying the outdoors and nature. |
| Positive | Clear | V | 12/05/2021 23:24 | Var í Öskjuhlíðinni að skoða stíga og fugla og kanínur með litlu í kvöld. Varð litið í átt til gossins, sá feikiháan gosstrók, skærappelsínugulan. Þetta er svo flott! | Was at Öskjuhlíð <park in Reykjavik> tonight with my little one exploring the trails, birds and bunnies. Looked up towards the eruption, saw an extremely high eruption fountain, bright orange. It was so beautiful! |
| Neutral | Ambiguous - could be positive | V | 14/04/2021 11:54 | Langar þig að skoða hvernig eldgosið við Fagradalsfjall hefur þróast og hraunið breitt úr sér?Náttúrufræðistofnun Íslands hefur í samstarfi við Háskóla | Would you like to see how the Fagradalsfjall eruption has evolved and the lava extended? The Iceland Institute of Natural History in collaboration with the University of Iceland, |

| | | | | Íslands, almannavarnir og Landmælingar Íslands útbúið glæsileg þrívíddarkort af því. | civil protection and National Land survey of Iceland has published a remarkable 3D map. |
|---|---|---|---|---|---|
| Negative | Ambiguous, use of humour. Negative sentiment addressed at the radio interviewee, rather than at the eruption | V | 08/09/2021 07:42 | Það er maður í útvarpinu mínu að kvarta yfir því að það hafi komið covid og eldgos því það hafið séð til þess að það væri ekki rædd pólitík í vor. | There is a man on the radio complaining that covid and the eruption prevented political debates this spring. |
| Negative | Ambiguous, use of sarcasm | V | 15/04/2021 18:36 | Þetta eldgos er nú orðið svoldið þreytt, er ekki hægt að fara fá eitthvað almennilegt fútt í þetta, loftstein eða eitthvað? | This eruption is becoming a bit tired, can't we get some proper excitement, like a meteorite or something? |
| Negative. | Ambiguous. Negative sentiment addressed to covid-19 restrictions. Labelled Negative | V | 13/04/2021 14:13 | Þúsund manns að horfa á eldgos og 100 manns í leikhúsi en enginn fær að horfa á kappleiki í sporti. Hvað er í gangi ??????????? | Thousand people watching an eruption and 100 people in the theatre, but noone is allowed to watch [live] sports matches. What is going on???????????? |

| Negative | Ambiguous, use of humour | V | 05/07/2021 23:14 | ekki farið og séð gosið, ekki farið í sky lagoon... instagram fer að loka á aðganginn minn 😵 😵. Spurning hvort ferðini í stuðlagil 2020 lengi aðeinsíðví | not been to see the eruption, not been to sky lagoon… instagram is going to be closing my instagram account 😵 😵. Wonder if my trip to Stuðlagil in 2020 is extending it a bit |
| Positive | Ambiguous. Positive sentiment refers to the prime minister, rather than the earthquake | E | 20/10/2020 14:10 | Forsætisráðherra heldur kúlinu þegar skjálftinn kemur í beinni. | The prime minister remains cool when the earthquake strikes during live interview. |

**Table A2: Examples of tweets from the dataset with keywords related to earthquakes (E), and the volcanic eruption (V). Column "Label" shows how the tweet was labelled in the dataset. Column "Label certainty" shows whether the expressed sentiment was straightforward ("clear") or "ambiguous" to label. Where the labelling was ambiguous an explanation is included. The table includes the original tweets in Icelandic, and our translation to English. The selected tweets are unchanged from the original posts with one exception. Some tweets included links to the original posts; we removed these to preserve the posters' anonymity.**

**Appendix A references**

Devlin, J., Chang, M.-W., Lee, K., and Toutanova, K. N.: BERT: Pre-training of Deep Bidirectional Transformers for Language Understanding, in: Proceedings of the 2019 Conference of the North American Chapter of the Association for Computational Linguistics: Human Languages Technologies, Minneapolis, Minnesota, 2019.

Geirhos, R., Jacobsen, J.-H., Michaelis, C., Zemel, R., Brendel, W., Bethge, M., and Wichmann, F. A.: Shortcut learning in deep neural networks, Nat Mach Intell, 2, 665–673, https://doi.org/10.1038/s42256-020-00257-z, 2020.

Ingólfsdóttir, S. L., Ragnarsson, P., Jónsson, H., Simonarson, H., Thorsteinsson, V., and Snæbjarnarson, V.: Byte-Level Grammatical Error Correction Using Synthetic and Curated Corpora, in: Proceedings of the 61st Annual Meeting of the Association for Computational Linguistics (Volume 1: Long Papers), ACL 2023, Toronto, Canada, 7299–7316, https://doi.org/10.18653/v1/2023.acl-long.402, 2023.

Kale, M. and Rastogi, A.: Text-to-Text Pre-Training for Data-to-Text Tasks, https://doi.org/10.48550/arXiv.2005.10433, 8 July 2021.

Maas, A. L., Daly, R. E., Pham, P. T., Huang, D., Ng, A. Y., and Potts, C.: Learning Word Vectors for Sentiment Analysis, ACL, 2019.

Mohammad, S., Kiritchenko, S., Sobhani, P., Zhu, X., and Cherry, C.: SemEval-2016 Task 6: Detecting Stance in Tweets, in: Proceedings of the 10th International Workshop on Semantic Evaluation (SemEval-2016), SemEval 2016, San Diego, California, 31–41, https://doi.org/10.18653/v1/S16-1003, 2016.

OpenAI, Achiam, J., Adler, S., Agarwal, S., Ahmad, L., Akkaya, I., Aleman, F. L., Almeida, D., Altenschmidt, J., Altman, S., Anadkat, S., Avila, R., Babuschkin, I., Balaji, S., Balcom, V., Baltescu, P., Bao, H., Bavarian, M., Belgum, J., Bello, I., Berdine, J., Bernadett-Shapiro, G., Berner, C., Bogdonoff, L., Boiko, O., Boyd, M., Brakman, A.-L., Brockman, G., Brooks, T., Brundage, M., Button, K., Cai, T., Campbell, R., Cann, A., Carey, B., Carlson, C., Carmichael, R., Chan, B., Chang, C., Chantzis, F., Chen, D., Chen, S., Chen, R., Chen, J., Chen, M., Chess, B., Cho, C., Chu, C., Chung, H. W., Cummings, D., Currier, J., Dai, Y., Decareaux, C., Degry, T., Deutsch, N., Deville, D., Dhar, A., Dohan, D., Dowling, S., Dunning, S., Ecoffet, A., Eleti, A., Eloundou, T., Farhi, D., Fedus, L., Felix, N., Fishman, S. P., Forte, J., Fulford, I., Gao, L., Georges, E., Gibson, C., Goel, V., Gogineni, T., Goh, G., Gontijo-Lopes, R., Gordon, J., Grafstein, M., Gray, S., Greene, R., Gross, J., Gu, S. S., Guo, Y., Hallacy, C., Han, J., Harris, J., He, Y., Heaton, M., Heidecke, J., Hesse, C., Hickey, A., Hickey, W., Hoeschele, P., Houghton, B., Hsu, K., Hu, S., Hu, X., Huizinga, J., Jain, S., et al.: GPT-4 Technical Report, https://doi.org/10.48550/arXiv.2303.08774, 4 March 2024.

Ott, M., Edunov, S., Baevski, A., Fan, A., Gross, S., Ng, N., Grangier, D., and Auli, M.: fairseq: A Fast, Extensible Toolkit for Sequence Modeling, in: Proceedings of the 2019 Conference of the North American Chapter of the Association for Computational Linguistics (Demonstrations), Minneapolis, Minnesota, 48–53, https://doi.org/10.18653/v1/N19-4009, 2019.

Ruder, S., Peters, M. E., Swayamdipta, S., and Wolf, T.: Transfer Learning in Natural Language Processing, in: Proceedings of the 2019 Conference of the North American Chapter of the Association for Computational Linguistics: Tutorials, Minneapolis, Minnesota, 15–18, https://doi.org/10.18653/v1/N19-5004, 2019.

Snæbjarnarson, V. and Einarsson, H.: Cross-Lingual QA as a Stepping Stone for Monolingual Open QA in Icelandic, https://doi.org/10.48550/arXiv.2207.01918, 5 July 2022.

Socher, R., Perelygin, A., Wu, J., Chuang, J., Manning, C. D., Ng, A., and Potts, C.: Recursive Deep Models for Semantic Compositionality Over a Sentiment Treebank, in: Proceedings of the 2013 Conference on Empirical Methods in Natural Language Processing, EMNLP 2013, Seattle, Washington, USA, 1631–1642, 2013.

Vaswani, A., Shazeer, N., Parmar, N., Uszkoreit, J., Jones, L., Gomez, A. N., Kaiser, Ł., and Polosukhin, I.: Attention is All you Need, in: Advances in Neural Information Processing Systems, 2017.

Wolf, T., Debut, L., Sanh, V., Chaumond, J., Delangue, C., Moi, A., Cistac, P., Rault, T., Louf, R., Funtowicz, M., Davison, J., Shleifer, S., von Platen, P., Ma, C., Jernite, Y., Plu, J., Xu, C., Le Scao, T., Gugger, S., Drame, M., Lhoest, Q., and Rush, A.: Transformers: State-of-the-Art Natural Language Processing, in: Proceedings of the 2020 Conference on Empirical Methods in Natural Language Processing: System Demonstrations, Online, 38–45, https://doi.org/10.18653/v1/2020.emnlp-demos.6, 2020.