# Peer review of "Brief communication: Small-scale geohazards cause significant and highly variable impacts on emotions"

_Natural Hazards and Earth System Sciences, 2023_

## Author Response (AR1)

**Reply to reviewers**

**AC1**: 'Reply on RC1', Evgenia Ilyinskaya, 27 Jun 2024   reply

The reviewer's comments are **highlighted in bold** in this response

**This paper is a well written focussed brief communications that needs only minor revision for publication.**

Reply: We thank the reviewer for the positive and thoughtful review

**My suggestions (in no particular order of importance) are as follows:**

- o **Please add more specific summary information in it. For example, how many tweets, examples of sentiments found, what was the measurable change in sentiments. It is currently very hard level and does not work well as a summary.**

Reply: We understand this comment to be addressed at the Abstract. We have added the number of tweets and the positive-to-negative sentiment ratio to the abstract, as suggested by the reviewer. Updated text: "The impact of geohazards on the mental health of the local populations is well recognised but understudied. We used natural language processing (NLP) of Twitter posts (n = 10,341) to analyse the sentiments expressed in relation to pre-eruptive seismic unrest and a subsequent volcanic eruption in Iceland 2019-2021. Despite the small size and negligible material damage, we show that these geohazards were associated with a measurable change in expressed emotions in the local populations. The seismic unrest was associated with predominantly negative sentiments (positive-to-negative sentiment ratio 1:1.3), but the eruption with predominantly positive (positive-to-negative sentiment ratio 1.4:1). We demonstrate a cost-effective tool for gauging public discourse that could be used in risk management. "

We could not add examples of sentiments to the abstract because of its 100-word limit. We have added examples of sentiments to the main text, the table is also attached here as a Supplement.

- o **This communications was submitted in 2023 but only one of the 12 references is from 2022 and one from 2021. This feels a bit dated, as this field has been moving on over the past few years. Introduced a few more recent examples of the literature surrounding social media and geohazards.**

Reply: We thank the reviewer for pointing this out. We have reviewed the recent progress in the field and have added the following publications to the manuscript, within the Introduction and elsewhere:

[revised manuscript text omitted]

- o **You might want to briefly mention the significance of your findings in the broader context of geohazard risk management and public health.**

Reply:

We have added the following discussion of broader context to the Introduction and the Conclusions:

Introduction added text: "The impacts of relatively small events are not well covered in literature, although they are much more common than large ones. Small eruptions (Volcanic Explosivity Index ≤2) account for ~80% of eruptions worldwide (Siebert et al., 2015). Pre-eruptive unrest is currently not considered in volcanic hazard and risk assessments beyond material damage to structures and has not been researched. There is a pressing need to understand the impacts of events of all sizes on the local populations, given that ~500 million people live within 50 km distance from a volcano, and rising numbers within 10 km (Freire et al., 2019)."

Conclusions, newly added text is within *asterisks*: "Our findings are important for risk assessment and management because we show that even small-sized geohazards without significant material damage can cause a measurable change in expressed sentiments in the local populations, which may indicate an impact on people's mental health. *Pre-eruptive unrest is currently somewhat of a 'forgotten' volcanic hazard, the public health impacts of which have not been researched. Furthermore, it is not included in educational and scientific resources such as the Encyclopedia of Volcanoes (Rymer, 2015) and the VolFilm series (Global Volcanism Program | Video Collections | VOLFilms Collection, 2024). We show that pre-eruptive events (here, the seismic unrest) can potentially be more detrimental to mental well-being than the actual eruption and should be considered in studies of impacts. In general, the impacts of living with uncertainty due to expectant but not yet materialised geohazards need further attention from the research and disaster risk reduction (DRR) communities.*

Incorporating sentiment analysis of crowd-sourced information, such as social media posts, into local risk management has the potential for immediate and longer-term benefits. While our method does not provide direct measures of the mental health state and impacts, and is not intended to replace more formal investigations, it may be used to quickly gauge whether communities are under stress and may require additional surveying and/or resources. Alternatively, knowing that the public views an eruption (or other natural phenomena) as a generally enjoyable and attractive event may

assist the risk managers in choosing the most effective approach to achieve compliance should they need to restrict the site access.

*Our study provides a valuable contribution to longitudinal studies of mental health impacts in the local populations. The volcano-seismic events on Reykjanes are still ongoing at the time of writing (June 2024) and are considered likely to continue for years or potentially decades. Long-duration, dynamically evolving volcanic events are common worldwide, for example at Kīlauea volcano in Hawaii.  The method we demonstrate allows the capture and analysis of sentiments experienced and expressed in situ, which cannot be reliably reproduced by retrospective investigations. The contemporary meaning of people's lived experiences is essential but ephemeral, and the ability to recall fades and changes over time. With the literal loss of familiar landscapes, and homes, people also lose connection to their memories (Árnason and Hafsteinsson, 2023). Further studies could focus on designing methodologies for quantifying emotional impacts, mental health outcomes and risk perception changes using social media data in the Reykjanes case study, and elsewhere.*

Finally, in our quest to reduce the risk posed by geohazards – in this case, a small eruption - we should not dismiss the potential mental health benefits from allowing people to experience them where possible, even though the benefits will be difficult to quantify and weigh up against the (often more obvious) risks."

- o **These are well written, succinct, and refer the reader to the appendix for further information.**

Reply: We thank the reviewer for the positive feedback.

- o **Results and Discussion.**
  - ▪ **I suggest you 'introduce' Figures 1 and 2 in terms of what is being shown, rather than just refer to results from them. Provide even 1-2 sentences pointing out to the reader what they are seeing. In Figure 1 is presented.... One can observe...**

Reply: Thank you for this suggestion. We have added the following explanation of  Figure 1 in the main text "Figure 1 is a time-series spanning our study period and shows changes in the number of tweets with earthquake-related and eruption-related keywords (from here on termed 'earthquake-tweets' and 'eruption-tweets', respectively), and their sentiment labelling. Figure 1a shows the time-dependent changes in number of tweets,

and the number of earthquakes by calendar week; Figure 1b shows the changes in number of tweets labelled as negative and positive, respectively, by week; and Figure 1c shows the changes in positive-to-negative sentiment ratio by week. "

and for Figure 2:

To analyse this relationship quantitatively, we plotted the number of earthquake-tweets as a function of the number of earthquakes, shown in Figure 2.

**See comments under Figure 1 and 2 below for other comments (that sometimes relate to explanations in the text).**

Reply: all comments are addressed below

- **I'd like a bit more depth about uncertainties and limitations (beyond the brief statement at end of methods). In particular, can you briefly discuss limitations of NLP in interpreting the nuances of human emotions, especially in languages with complex idiomatic expressions like Icelandic.**

Reply: We have added the following discussion to the revised version: "The representations one can extract from pre-trained foundation models are limited to the relations that can be inferred from their training data. If these do not sufficiently cover the style or terminology one is interested in, then the representations will not be robust. This can be particularly pronounced in the case of idiomatic expressions for low-resource languages when using multilingual models. As we target Icelandic, we use a pre-trained model trained on only Icelandic and English to mitigate these issues. Furthermore, fine-tuning these models using the labelled data can be seen as a way to improve these representations and their decision boundaries. Finally, as improved models and datasets become available, we can expect higher accuracy and lower uncertainty when these are applied to the data we have collected."

- **Including specific tweet examples as evidence of these sentiment changes could help the reader's understanding. I realize that this is a short communications but I was left without a good feeling for what was categorised positive vs. negative and whether you had categories within them (and neutral) or 'more' or 'less' positive/negative/neutral.**

Reply: We have added a table with examples to the Appendix (Table A2), and discussion of it in the text. We selected 15 tweets representing eruption- and earthquake-related keywords and the 3 labelling categories (neutral, positive,

negative). Table A2 also shows whether the expressed sentiment was straightforward or ambiguous to label. In the case where the labelling was ambiguous, an explanation is included in the table. Humour and sarcasm were common reasons for labelling ambiguous sentiments. The table is included here as a Supplement attachment.

- o **Further elaboration on how your insights could be integrated into existing geohazard management frameworks might enhance the paper's applicability.**

Reply: We believe this point has been addressed in our reply to a previous comment ("You might want to briefly mention the significance of your findings in the broader context of geohazard risk management and public health.")

- o **Figure 1.**
  - ▪ **Overall a really nice figure.**

Reply: Thank you

- o **Suggest that blue and red for Fig. 1C might be difficult for colour blind people unless you add shading.**

Reply: We have checked color blindness guidelines and ran the figure through a simulator tool (https://www.color-blindness.com/coblis-color-blindness-simulator/) and found that red/blue combination in our figure should not cause problems

- o **I'm not getting the 'yellow' refferred to in the figure caption— do you mean orange? This might be an issue of the PDF.**

Reply: We agree that the colour can be interpreted as orange. We will refer to it as orange in the final version

- o **In figure caption reiterate how many total tweets there are.**
- o **In Figure 1a, earthquakes from what minimum to maximum magnitude?**
- o **Refer reader back to Appendix A for the detailed list of keywords.**

Reply to the 3 comments above: We have modified Figure 1 caption according to the reviewer's suggestions: "Figure 1: Time series plot of Twitter data (n of Tweets = 10,341) during the pre-eruptive seismic unrest period (December 2019 – March 2021) and the eruption period (March – September 2021, highlighted with orange). The tweets were selected by a set of earthquake- and eruption-related keywords in Icelandic, as listed in Appendix A.  a) The weekly number of earthquake- and eruption-related tweets and the

weekly number of earthquakes. The number of weekly earthquakes is shown separately for the Reykjanes peninsula and Iceland. The magnitude of earthquakes ranged from Mw 0 to 5.6. b) The weekly number of tweets evaluated as expressing positive ('pos') or negative ('neg') sentiments by the NLP model. c) The average positive/negative ('pos/neg') weekly ratio, as evaluated by the NLP model in the whole dataset and manually in the data subset. The data shown in c) include only weeks where the total number of tweets was > 10 to avoid bias introduced by very low numbers"

- **Figure 2.**
  - **This figure does not work as well.**
  - **See comments on Figure 1 re earthquake magnitude and labelling.**
  - **I'd be curious to see this figure log-log, given the clustering of values in the lower decades of n. Consider doing a four-part figure rather than two part showing the y-axis linear and log and x-axis always log.**
  - **I suggest you have two different variables for x- and y-axis (not n for each, that is confusing).**

Reply to all comments related to Figure 2:

Figure 2 was updated to be 4-panel as suggested by the reviewer (see revised manuscript file).

We have modified Figure 2 caption based on relevant comments from Figure 1, i.e. added the earthquake magnitude:

"Figure 2: Scatter plot showing the number of earthquake-related tweets as a function of the number of earthquakes each week over the study period (December 2019 - December 2021). The solid line is a linear regression model fit, and the dotted lines are the 5% significance level. Panel la) displays earthquakes on the Reykjanes peninsula (magnitude range Mw 0 – 5.6). The r2 between the number of earthquake-related tweets and number of earthquakes is 0.69. Panel b) shows the same data as a) but on a log-log axis. Panel c) displays earthquakes in all of Iceland, excluding the Reykjanes peninsula (magnitude range Mw 0 – 4.8). There is no statistically significant relationship between the two variables. Panel d) shows the same data as c) but on a log-log axis"

We politely disagree with the reviewer's comment that plotting the number of earthquakes against the number of tweets is confusing. Since the reviewer did not suggest a better alternative, we are not sure what one would be. We

have rephrased the axis titles and the caption (see new version above) to hopefully improve clarity.

- ○ **Appendix A.**

    - ▪ **This feels (like introduction) a tad dated for references on previous work.**

Reply: We have updated references w.r.t. language models and surveys on the use of sentiment analysis, such as

Venkit, P., Srinath, M., Gautam, S., Venkatraman, S., Gupta, V., Passonneau, R., and Wilson, S.: The Sentiment Problem: A Critical Survey towards Deconstructing Sentiment Analysis, in: Proceedings of the 2023 Conference on Empirical Methods in Natural Language Processing, EMNLP 2023, Singapore, 13743–13763, https://doi.org/10.18653/v1/2023.emnlp-main.848, 2023.

He, P., Gao, J., and Chen, W.: DeBERTaV3: Improving DeBERTa using ELECTRA-Style Pre-Training with Gradient-Disentangled Embedding Sharing, The Eleventh International Conference on Learning Representations, 2022.

Ingólfsdóttir, S. L., Ragnarsson, P., Jónsson, H., Simonarson, H., Thorsteinsson, V., and Snæbjarnarson, V.: Byte-Level Grammatical Error Correction Using Synthetic and Curated Corpora, in: Proceedings of the 61st Annual Meeting of the Association for Computational Linguistics (Volume 1: Long Papers), ACL 2023, Toronto, Canada, 7299–7316, https://doi.org/10.18653/v1/2023.acl-long.402, 2023.

- ○ **Stating 'in recent years' but citing a reference from 2017 does not work well.**

Reply: Thank you, we have rephrased this. See also our reply regarding adding more recent references

- ○ **For keywords, how were plural, and slight spelling errors deal with? If NLP dealt with these, then state that.**

Reply: We have added this clarification to the updated Appendix version. "We set the keywords to include all correct grammatical variations of 'eruption' and 'earthquake' in Icelandic, including the plural case. Spelling errors were not dealt with. This could be addressed with, e.g. Levenshtein distance, but there are common words in Icelandic that have only small spelling differences with "gos" (eruption) (e.g. "gas"), which would have increased the number of false positives. Another approach could be to run the text through

a grammatical error correction system before annotation. However, these have not been rigorously tested on non-editorial text such as tweets (Ingólfsdóttir et al. 2023 https://doi.org/10.18653/v1/2023.acl-long.402). We do not use these extra steps since the main keywords, "gos"and "skjálfti" (eruption and earthquake), and their grammatical variations are short and easy-to-spell words in Icelandic."

- o **For which were sentiments were negative, positive and netural, it would be nice to show a table of examples.**

Reply: See the reply to the same comment further up. We have added a Table to the Appendix with these examples, and it is attached here as a Supplement.
* * *
AC2: ['Reply on RC2'](), Evgenia Ilyinskaya, 27 Jun 2024  reply
The reviewer's comments are **highlighted in bold.**

**The manuscript presents an innovative study on the impact of small-scale geohazards, specifically pre-eruptive seismic unrest and a volcanic eruption in Iceland in 2021, on the local population's expressed sentiments via social media, utilizing natural language processing (NLP) techniques. The study's novelty lies in using social media data to evaluate the emotional and mental health impacts of geohazards, offering insights that could be instrumental in risk management and emergency response planning. The methodological approach, combining manual and AI-assisted sentiment analysis, is particularly commendable for its attempt to navigate the challenges of language specificity and context sensitivity in sentiment analysis. I recommend this manuscript for publication following a 'minor revision' to address the points raised.**

**Strengths of the manuscript:**

- **Leveraging NLP to analyze social media data for sentiment analysis related to geohazards is innovative and provides a scalable method for real-time sentiment tracking.**

- **The discovery that small-scale geohazards can cause significant emotional impacts, with a distinction between negative sentiments during seismic unrest and positive sentiments during the eruption phase, is an important contribution to both geohazard risk management and mental health fields.**

- **The detailed description of the methodology, including the adaptation of the model to handle Icelandic sentiment analysis and the efforts to**

**mitigate bias and inaccuracies, showcases a commendable level of methodological rigor.**

**Areas for Improvement**

- **The study's reliance on Twitter data, while innovative, raises questions about the representativeness of the findings. With Twitter's user demographic not fully representing the entire population, future research could benefit from incorporating data from multiple social media platforms to capture a wider range of sentiments. I suggest that the authors explicitly state this limitation.**

Reply: We fully agree. The original version already had the following text in the Methods section: "The main potential limitation of our approach is that views expressed on Twitter may not fully represent views of people who chose to use different social media platforms, or none at all; this can be explored in future research by including more than one social media platform. This would demonstrate the applicability of the method to other countries, where popularity of different social media platforms may differ." We believe this text addresses the reviewer's comment, but to make it more explicit to the paper's readers, we have moved it to a more prominent place (now in the Methods summary rather than in one of the Methods subsections).

- **While the adaptation of the model for Icelandic is a strength, the reliance on a model initially trained on English data and the challenges associated with keyword masking deserve further discussion. The implications of these methodological choices on the findings' accuracy and generalizability should be addressed more thoroughly.**

Reply: The model used was pre-trained on both Icelandic and English. Then, we initially adapted the model using English-only sentiment data, a method shown to benefit other languages (Conneau et al. 2020, Pires et al. 2019).

Alexis Conneau, Kartikay Khandelwal, Naman Goyal, Vishrav Chaudhary, Guillaume Wenzek, Francisco Guzmán, Edouard Grave, Myle Ott, Luke Zettlemoyer, and Veselin Stoyanov. 2020. Unsupervised Cross-lingual Representation Learning at Scale. In *Proceedings of the 58th Annual Meeting of the Association for Computational Linguistics*, pages 8440–8451, Online. Association for Computational Linguistics.

Telmo Pires, Eva Schlinger, and Dan Garrette. 2019. How Multilingual is Multilingual BERT?. In *Proceedings of the 57th Annual Meeting of the Association for Computational Linguistics*, pages 4996–5001, Florence, Italy. Association for Computational

Linguistics.

Finally, we trained on Icelandic-only sentiment data. We acknowledge other ways to set up such a model but emphasise that our goal in this work is not to exhaustively compare those but simply evaluate one such methodology for the events we cover. This is a standard approach in NLP when data is used for transfer learning between a high-resource language and a low-resource language (Pfeiffer et al. 2020; Snæbjarnarson et al. 2023).

Jonas Pfeiffer, Ivan Vulić, Iryna Gurevych, and Sebastian Ruder. 2020. MAD-X: An Adapter-Based Framework for Multi-Task Cross-Lingual Transfer. In *Proceedings of the 2020 Conference on Empirical Methods in Natural Language Processing (EMNLP)*, pages 7654–7673, Online. Association for Computational Linguistics.

Vésteinn Snæbjarnarson, Annika Simonsen, Goran Glavaš, and Ivan Vulić. 2023. Transfer to a Low-Resource Language via Close Relatives: The Case Study on Faroese. In Proceedings of the 24th Nordic Conference on Computational Linguistics (NoDaLiDa), pages 728–737, Tórshavn, Faroe Islands. University of Tartu Library.

Keyword masking was used to remove easy queues from the dataset and mitigate the pre-trained model's inductive biases. We hypothesise that this also makes the model more robust and gives it a fairer assessment when measured over the evaluation dataset.

If the method is to be followed for other languages, a limiting factor is that the mentioned resources need to be present in some form to cover a given language, i.e., pre-training data or pre-trained models, along with sufficient labelled sentiment data. In these cases, recent advances in general-purpose large-language models (see, e.g. Achiam et al. 2023) may be of value in both direct evaluation or pseudo-labeling training data in low-resource settings.

Achiam, Josh, Steven Adler, Sandhini Agarwal, Lama Ahmad, Ilge Akkaya, Florencia Leoni Aleman, Diogo Almeida et al. "Gpt-4 technical report." arXiv preprint arXiv:2303.08774 (2023).

We have added a discussion of these points to the revised version.

- **The study establishes that small-scale geohazards have significant emotional impacts but does not quantify these impacts in a way that could be useful for risk management or emergency response planning. Future studies could explore methodologies for quantifying emotional impacts regarding mental health outcomes or risk perception changes.**

Reply: We fully agree with the reviewer. We have added new text (see below) that addresses this comment as far as is feasible within the scope of a Brief Communication manuscript. We cannot quantify impacts with the available dataset and have stated that this should form part of future studies, as suggested by the reviewer.

We have added the following discussion of broader context to the Introduction and the Conclusions:

Introduction added text: "The impacts of relatively small events are not well covered in literature, although they are much more common than large ones. Small eruptions (Volcanic Explosivity Index ≤2) account for ~80% of eruptions worldwide (Siebert et al., 2015). Pre-eruptive unrest is currently not considered in volcanic hazard and risk assessments beyond material damage to structures and has not been researched. There is a pressing need to understand the impacts of events of all sizes on the local populations, given that ~500 million people live within 50 km distance from a volcano, and rising numbers within 10 km (Freire et al., 2019)."

Conclusions, newly added text is within *asterisks*: "Our findings are important for risk assessment and management because we show that even small-sized geohazards without significant material damage can cause a measurable change in expressed sentiments in the local populations, which may indicate an impact on people's mental health. *Pre-eruptive unrest is currently somewhat of a 'forgotten' volcanic hazard, the public health impacts of which have not been researched. Furthermore, it is not included in educational and scientific resources such as the Encyclopedia of Volcanoes (Rymer, 2015) and the VolFilm series (Global Volcanism Program | Video Collections | VOLFilms Collection, 2024). We show that pre-eruptive events (here, the seismic unrest) can potentially be more detrimental to mental well-being than the actual eruption and should be considered in studies of impacts. In general, the impacts of living with uncertainty due to expectant but not yet materialised geohazards need further attention from the research and disaster risk reduction (DRR) communities.*

Incorporating sentiment analysis of crowd-sourced information, such as social media posts, into local risk management has the potential for immediate and longer-term benefits. While our method does not provide direct measures of the

mental health state and impacts, and is not intended to replace more formal investigations, it may be used to quickly gauge whether communities are under stress and may require additional surveying and/or resources. Alternatively, knowing that the public views an eruption (or other natural phenomena) as a generally enjoyable and attractive event may assist the risk managers in choosing the most effective approach to achieve compliance should they need to restrict the site access.

*Our study provides a valuable contribution to longitudinal studies of mental health impacts in the local populations. The volcano-seismic events on Reykjanes are still ongoing at the time of writing (June 2024) and are considered likely to continue for years or potentially decades. Long-duration, dynamically evolving volcanic events are common worldwide, for example at Kīlauea volcano in Hawaii. The method we demonstrate allows the capture and analysis of sentiments experienced and expressed in situ, which cannot be reliably reproduced by retrospective investigations. The contemporary meaning of people's lived experiences is essential but ephemeral, and the ability to recall fades and changes over time. With the literal loss of familiar landscapes, and homes, people also lose connection to their memories (Árnason and Hafsteinsson, 2023). Further studies could focus on designing methodologies for quantifying emotional impacts, mental health outcomes and risk perception changes using social media data in the Reykjanes case study, and elsewhere.*

Finally, in our quest to reduce the risk posed by geohazards – in this case, a small eruption - we should not dismiss the potential mental health benefits from allowing people to experience them where possible, even though the benefits will be difficult to quantify and weigh up against the (often more obvious) risks."

- **A deeper discussion on the limitations of using an NLP model trained primarily on English data for analyzing Icelandic tweets, including any potential biases or inaccuracies introduced, would strengthen the study.**

Reply: See response to a previous comment. The model was pre-trained on both Icelandic and English. The final step of training was done on Icelandic only.

- **Developing and discussing methods for quantifying the emotional and mental health impacts of geohazards could significantly enhance the practical implications of the research.**

- **While the study focuses on immediate sentiments expressed during the geohazards, a discussion on the potential long-term mental health impacts would provide a more comprehensive view of the geohazards' effects.**

Reply to the 2 comments above: Similar to a previous comment, while we fully agree with the reviewer, we do not think this is feasible in this Brief Communication. The pathogenesis of mental health impacts is a complex and poorly understood field, and any discussion of the potential outcomes would not be appropriate with the limitations of the dataset presented in this study. We have made additions to the text (see reply to a previous comment) that expand on the broader context and future directions, and hope that the reviewer and the editor will find these acceptable.

**This manuscript provides valuable insights into the emotional impacts of geohazards on local populations and introduces a novel methodological approach to sentiment analysis in this context. With improvements in representativeness, model transparency, and impact quantification, this work could significantly contribute to the fields of geohazard risk management and mental health. I recommend this manuscript for publication following a 'minor revision' to address the points raised.**

Reply: We thank the reviewer for the detailed and constructive comments